# HeaP: Hierarchical Policies for Web Actions using LLMs

## Abstract

Large language models (LLMs) have demonstrated remarkable capabilities in performing a range of instruction following tasks in few and zero-shot settings. However, teaching LLMs to perform tasks on the web presents fundamental challenges – combinatorially large open-world tasks and variations across web interfaces. We tackle these challenges by leveraging LLMs to decompose web tasks into a collection of sub-tasks, each of which can be solved by a low-level, closed-loop policy. These policies constitute a *shared grammar* across tasks, i.e., new web tasks can be expressed as a composition of these policies. We propose a novel framework, Hierarchical Policies for Web Actions using LLMs (`HeaP`), that learns a set of hierarchical LLM prompts from demonstrations for planning high-level tasks and executing them via a sequence of low-level policies. We evaluate `HeaP` against a range of baselines on a suite of web tasks, including MiniWoB++, WebArena, a mock airline CRM, as well as live website interactions, and show that it is able to outperform prior works using orders of magnitude less data.

## 1 Introduction

Recent advances in instruction following large language models (LLMs) (Ouyang et al., 2022; Touvron et al., 2023) have shown impressive zero and few-shot capabilities in solving tasks by parsing natural language instructions and breaking them down into actionable steps (Yao et al., 2022b; Huang et al., 2022b; Ahn et al., 2022). In this paper, we focus on the problem of teaching LLMs to perform tasks on the web, for instance booking flights or making appointments. Assisting humans in performing web tasks has significant implications on a variety of domains given the pervasive nature of web and cloud-based applications in everyday life.

Prior works collect large amounts of demonstrations of web tasks to train language models (Furuta et al., 2023; Gur et al., 2022; Humphreys et al., 2022; Liu et al., 2018; Shi et al., 2017). However, teaching LLMs to perform tasks on the web presents fundamental challenges. (1) *Combinatorially large open-world tasks*: There are countless ways to interact with the web, leading to a combinatorially large space of tasks such as booking flights, making appointments, payments, etc. (2) *Variations across web interfaces*: Web interfaces differ from one website to another, e.g. booking a flight on JetBlue is different from booking it on United. Hence, it is intractable to cover all such variations in tasks and interfaces in the training data, and have a single supervised model that can solve all tasks.

Our key insight is to leverage LLMs to *decompose* complex web tasks into a set of modular sub-tasks, each of which can be solved by a low-level, closed-loop web policy. These policies constitute a *shared grammar* across tasks, i.e., any new web task can be expressed as a composition of these policies. For example, the task of booking a flight can be expressed as a sequence of policies for filling source and destination airports, choosing flight dates, and filling in passenger details. Each low-level policy is specialized for a particular sub-task, e.g. a fill text policy can work on text boxes across web user interfaces (UIs) that either require clicking and typing text, or require typing partial text and auto-completing from a list of options.

While manually programming these policies can be tedious, it is much easier to learn them from humans performing varied tasks on the web. We propose a novel framework, **H**ierarchical **P**olicies for Web Actions using LLMs (`HeaP`), that learns a set of hierarchical LLM prompts for planning

high-level tasks and executing low-level policies. We first collect raw demonstrations from a human user, auto-label them with low-level policies, and auto-generate both task and policy prompts. At inference time, given a task objective, we hierarchically invoke an LLM to first generate a task plan and then generate actions for each policy in the plan. `HeaP` enables LLMs to respond effectively to dynamic web pages as well as generalize across tasks and interfaces from few-shot demonstrations.

Experimentally, we evaluate `HeaP` on a range of increasingly complex benchmarks: MiniWoB++, WebArena, a mock airline CRM simulator and live website interactions. We show that `HeaP` has significantly better task success rates and requires orders of magnitude less training (or demonstration) examples relative to prior work (see Table 1 for summary). We will open-source the code, simulator, and data at `https://anonymized_url`.

## 2 RELATED WORK

**Language models for web tasks.** Early work mapping natural language instructions into actions (Branavan et al., 2009; Artzi & Zettlemoyer, 2013; Diaz et al., 2013) has rapidly evolved resulting in new applications and datasets (Zhou et al., 2023; Deng et al., 2023). In language models performing web tasks, there are broadly three classes of methods: *(1) Reinforcement learning (RL) for web navigation* that train RL agents to navigate web interfaces (Humphreys et al., 2022; Li et al., 2020; Liu et al., 2018; Pasupat et al., 2018; Shi et al., 2017; Branavan et al., 2009; Gur et al., 2021). However, these are often sample inefficient and exploration on live websites can pose practical safety concerns. *(2) In-context learning with large language models* uses a combination of instructions and in-context examples with large language models (OpenAI, 2023a; Significant Gravitas, 2023; Wang et al., 2023; Friedman, 2022; LangChain, 2023), with a significant portion being open-source initiatives. However, they often rely on manually crafted prompts and heuristic strategies to tackle context lengths and task generalization, making it challenging to build on existing findings. *(3) Fine-tuning language models for web tasks* focuses on fine-tuning language models on specific web tasks and has emerged as a predominant approach in prior works (Gur et al., 2022; Furuta et al., 2023; Yao et al., 2022a; Gur et al., 2023; Brown et al., 2020; Gur et al., 2018; Xu et al., 2021; Mazumder & Riva, 2020). However, training such models has limitations such as an inability to generalize from few examples of tasks and interfaces, necessitating frequent retraining. As our method, `HeaP`, is compositional in how it uses the LLM, it is inherently not task-specific and does not have these shortcomings.

**Language models for decision making.** Instruction following large language models have shown impressive out-of-the-box decision making capabilities (Huang et al., 2022a; Brown et al., 2020; Radford et al., 2019; Huang et al., 2022b). This arises from an ability to break down complex tasks into smaller sub-tasks (Huang et al., 2022a; Zhou et al., 2021), reason about intermediate steps (Yao et al., 2022b; Wei et al., 2022), and recover from errors (Miao et al., 2023). As a results, LLMs in recent times, have found applications in diverse domains like web retrieval (Liu et al., 2023; Zaheer et al., 2022; Nakano et al., 2021; Schick et al., 2023; Nogueira & Cho, 2016), robotics (Ahn et al., 2022; Huang et al., 2022b), and text-based games (Yao et al., 2022b; 2020; Shridhar et al., 2020). Moreover, advances in multi-modal LLMs enable decision making from both language and image feedback (Shaw et al., 2023; Lee et al., 2023; Burns et al., 2022). However, such decision making capabilities remain to be explored for general purpose web tasks involving clicks, types, form filling, etc. Our approach, `HeaP`, leverages the task decomposition and reasoning capabilities of LLMs to perform a wide range of web tasks. With only a handful of examples, `HeaP` can generalize, showing improved performance over prior works (Liu et al., 2018; Gur et al., 2022; Humphreys et al., 2022; Furuta et al., 2023) that train specialized models with orders of magnitude more data.

## 3 PROBLEM FORMULATION

The overall goal is to learn a policy that performs a web task. The web task is represented as a context $\phi$, that can be (a) an explicit instruction such as *"Book me a flight from NYC to BOS"* (b) a structured dictionary defining the parameters of the task, or (c) a supporting set of texts such as a conversation where the instruction is implicit. Given the current context $\phi$, the goal is to perform a web task that achieves the task objective. We formulate this as a Contextual Markov Decision Process (CMDP), $< \Phi, \mathcal{S}, \mathcal{A}, \mathcal{T}, r >$, defined below:

- **Context,** $\phi \in \Phi$ is the web task objective expressed explicitly as an instruction or structured parameters or implicitly as a conversation

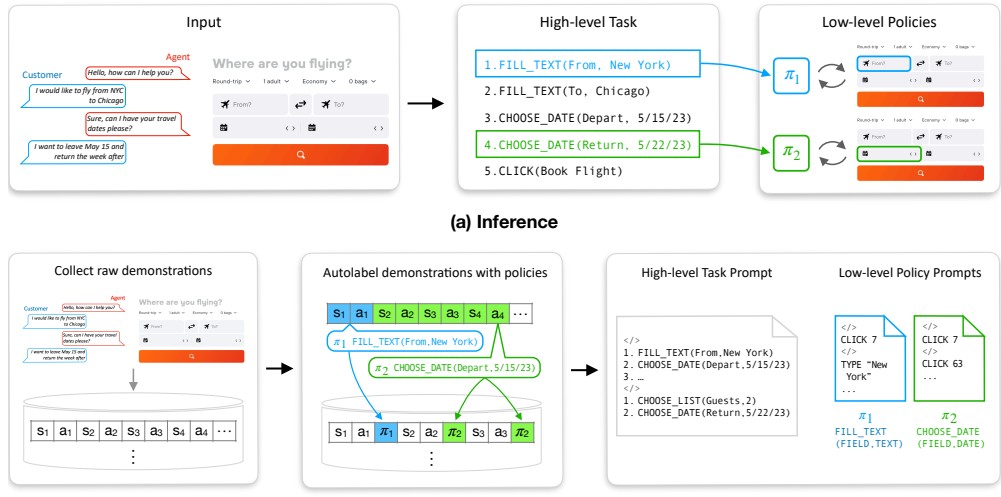

Figure 1: `HeaP` Overview: **(a) Inference:** High-level task planner creates a sequence of steps like filling text or choosing dates from an input context and starting webpage. Each step is a call to a low-level web policy that directly interacts with the webpage. (b) **Prompt Generation:** Dataset of raw state-action demonstrations is transformed into task and policy base prompts by first auto-labeling with policies and then generating prompts.

- **State,** $s \in \mathcal{S}$ is the current state of the webpage, i.e., the current DOM $d$ serialized as text. [1]
- **Action,** $a \in \mathcal{A}(s)$ are the set of web actions that can be performed on the current webpage, i.e. `click(id)`, `type(id,value)`, where `id` specifies an element in the webpage, and `value` is a string. The action space can be quite large since a typical webpage can have hundreds of elements, and `value` can be any arbitrary text.
- **Transition function,** $\mathcal{T}(s'|s, a)$ represents the change in the webpage on performing an action.
- **Reward,** $r(s, a)$ is awarded for reaching a set of subgoals, e.g. cancelling a flight has subgoals like finding the booking and then canceling it.

The goal is to learn a policy $\pi : \mathcal{S} \times \phi \to \mathcal{A}$ that maximizes performance, i.e., the cumulative reward $J(\pi) = \mathbb{E}_\pi \left[ \sum_{t=1}^{T} [r(s_t, a_t)] \right]$. Instead of explicitly defining the reward function and solving the MDP, we aim to learn this policy $\pi$ from demonstrations $\mathcal{D} = \{ (\phi^i, s_1^i, a_1^i, s_2^i, a_2^i, \dots ) \}_{i=1}^{N}$.

We leverage LLMs that are highly effective at generalizing from few-shot demonstrations without the need for fine-tuning. To do so, we translate demonstrations $\mathcal{D}$ into in-context examples for an LLM prompt $\mathcal{P}$. A simple way to do this is to flatten all demonstrations $\mathcal{D}$, i.e., concatenate the conversation $\phi$, with state action trajectories, and merge them together. However, a typical demonstration may consist of a lengthy chain of actions, with each state in the chain being the entire webpage document object model (DOM). In terms of total tokens, $N$ demonstrations each of $T$ timesteps, each step comprising of $X$ tokens of both conversation and webpage would result in $N \times T \times X$ tokens. This can quickly exhaust context space even for simple websites. We tackle this problem in our approach by hierarchically composing prompts.

## 4 APPROACH

We present a framework, **H**ierarchical **P**olicies for Web Actions using LLMs (`HeaP`), that performs a range of web tasks from natural language conversations by hierarchically invoking a Large Language Model (LLM). The framework consists of a hierarchy of two levels: a *high-level task planner* that in turns invokes a sequence of *low-level web policies*.

Consider the example in Fig. 1. Given a conversation with a customer looking to book flights, and a booking website, the task planner generates a plan, i.e, a sequence of steps to execute. Examples of steps are either filling a text box, choosing a date, or choosing an option from a drop-down. Each of these steps can be delegated to a corresponding web policy that interacts with the web page and

---

[1]For some tasks, the current webpage may not be sufficient to define state. In such cases, we can extend state to a history of previous webpages and actions.

---

**Algorithm 1** HeaP Inference: Compose policies to solve the task

1: **Input:** Context $\phi$, Current Webpage State $s_0$, LLM, Environment $\mathcal{T}$
2: $\xi \leftarrow$ TASKPLANNER$(\phi, s_0)$  ▷ Get task plan, i.e., sequence of calls to web policies
3: **for** $(\pi_i, \psi_i) \in \xi$ **do**
4:  WEBPOLICY$(\pi_i, \psi_i)$  ▷ Execute web policy
5: **end for**
6: **function** TASKPLANNER(Context $\phi$, State $s$)
7:  $\mathcal{P}_{\text{task}} \leftarrow$ Load base prompt for task planner
8:  $\xi \leftarrow$ LLM$(\phi, s, \mathcal{P}_{\text{task}})$  ▷ Predict plan given context, state
9:  **return** Plan $\xi = \{(\pi_1, \psi_1), (\pi_2, \psi_2), \ldots, (\pi_N, \psi_N)\}$
10: **end function**
11: **function** WEBPOLICY(Policy $\pi$, Instruction $\psi$)
12:  $\mathcal{P}_{\text{policy}} \leftarrow$ Load base prompt for web policy $\pi$
13:  $s \leftarrow$ GETCURRENTSTATE$()$, $a \leftarrow$ None, $a_{prev} \leftarrow \{\}$  ▷ Initialize state, action, prev actions
14:  **while** $a$ **not** done **do**
15:   $a \leftarrow$ LLM$(\psi, s, a_{prev}, \mathcal{P}_{\text{policy}})$  ▷ Predict action given instruction, state, prev actions
16:   $a_{prev} \leftarrow a_{prev} \cup a$  ▷ Append action to prev actions
17:   $s \leftarrow \mathcal{T}(s, a)$  ▷ Execute action to transition to next state
18:  **end while**
19: **end function**

---

executes web actions like clicking and typing. For instance, the Fill_TEXT(field, text) web policy searches for the web element corresponding to field, clicking it, typing a text and optionally choosing from a list of autocomplete options. On the other hand, the CHOOSE_DATE(field, date) web policy clicks on the web element, navigates a grid of dates and clicks on the correct date.

## 4.1 INFERENCE TIME: COMPOSE POLICIES TO SOLVE THE TASK

Algorithm 1 describes the inference time procedure. We take as input a context $\phi$, which can be a conversation or an explicit objective, and the current webpage state $s_0$. This is sent to a task planner that generates a plan. The plan is a sequence of calls to low-level web policies. Each element of the sequence is represented as a web policy type $\pi$ and instruction to the policy $\psi$, i.e., $\xi = \{(\pi_1, \psi_1), (\pi_2, \psi_2), \ldots (\pi_N, \psi_N)\}$. For example, CHOOSE_DATE(field, date) corresponds to calls to policy $\pi =$ CHOOSE_DATE with instruction $\psi =$ (field, date).

The web policies in plan $\xi$ are invoked one by one. Each policy $\pi_i$ predicts the next action $a$ given its instruction $\psi_i$, current state $s$, and previous actions $a_{prev}$. Once the policy issues the special action "DONE", control is handed back to the outer loop and the next policy is executed. When all policies in the plan $\xi$ are done, the task planner is invoked again for the next plan. The process is terminated when the task planner produces an empty plan.

Both the task planner and the web policies are calls to an LLM with different base prompts. The base prompt for the task planner shows examples of {input: [overall context $\phi$, current state $s_0$], output: plan $\xi$}. The base prompt for web policies shows examples of {input: [instruction $\psi_t$, current state $s_t$, previous actions $a_{1:t-1}$], output: next action $a_t$}. We additionally include chain-of-thought (CoT) reasoning Wei et al. (2022) to both task and policy prompts that forces the LLM to generate a series of short sentences justifying the actions it predicts. We found this to uniformly improve performance (Appendix B).

## 4.2 GENERATE TASK AND POLICY PROMPTS FROM DEMONSTRATIONS

To generate prompts from demonstrations, we collect demonstrations from human users performing tasks on the browser. We design a browser plugin to record webpage DOM $d$ and events such as clicks and types. Each demonstration is expressed as text by converting the DOM tree into a list of salient web elements like links, buttons, inputs. The parsed demonstration dataset is represented as $\mathcal{D} = \{(\phi, s_1, a_1, \ldots, s_T, a_T)\}$.

We then *autolabel* each step $t$ with a low-level policy $\pi_t$ and instruction $\psi_t$ to create a labeled dataset $\mathcal{D}_{label} = \{(\phi, s_1, a_1, (\pi_1, \psi_1), \ldots, s_T, a_T, (\pi_T, \psi_T))\}$. We leverage LLMs to autolabel demonstrations and describe details in Appendix. D. Finally, we convert demonstrations to base prompts for both high-level planner and low-level policies and list representative prompts in Appendix. G.

## 5 EXPERIMENTS

### 5.1 EXPERIMENTAL SETUP

**Environments.** We evaluate across 4 distinct environments, each emphasizing different components:

- **MiniWoB++** (Liu et al., 2018): An extension of the OpenAI MiniWoB benchmark covering a range of web interaction environments like form filling, search, choose dates, etc. We evaluate across 45 distinct tasks that don't rely on visual reasoning, and average over 50 seeds per task.
- **WebArena** (Zhou et al., 2023): A recent community benchmark offering complex web tasks across multiple domains. Compared to MiniWoB++, WebArena websites are highly realistic with tasks mirroring those that humans routinely perform on the internet. We evaluate on a set of 125 examples sampled from 12 distinct intents from 2 domains, Gitlab and OpenStreetMaps.
- **Airline CRM**: A new CRM simulator that we developed, modeled after customer service workflows of popular airline websites. Compared to MiniWoB++, Airline CRM offers longer-horizon tasks tied to a mock database, capturing typical CRM activities more effectively. We evaluate across 5 distinct tasks each with 20 randomized scenarios. More simulator details in Appendix E.
- **Live Websites**: Finally, we create an environment to interact with live websites, such as popular airlines like JetBlue, American, United. The raw browser content is considerably more complex, being ~100x larger than the simulators. We evaluate generalization across UIs by performing the same search-flight task across 3 very different website UIs and average across 10 runs per UI.

**Baselines.** We compare against various baselines including prior state-of-the-art (Furuta et al., 2023; Gur et al., 2022; Humphreys et al., 2022; Liu et al., 2018) and methods `Flat Zero-shot`, `Flat Few-shot`, `HeaP Zero-shot`, `HeaP Few-shot`. `Flat Zero-shot` is a single prompt containing only the instructions and no in-context examples. `Flat Few-shot` includes both instructions and in-context examples. Both of these follow a chain-of-thought prompting style similar to ReAct (Yao et al., 2022b). `HeaP Few-shot` and `HeaP Zero-shot` is our hierarchical prompting approach, `HeaP`, with and without in-context examples respectively. Detailed prompts for the different methods can be found in Appendix G. All 4 methods use the instruction fine-tuned `text-davinci-003`[2] model. We found it to perform better at multi-step reasoning compared to `gpt-3.5-turbo`[1](Ouyang et al., 2022) while being more cost-effective than `gpt-4`[1](OpenAI, 2023b). More details on model hyper-parameters in Appendix C.2.

**Metrics.** We define 3 metrics: Success Rate (%suc↑), Task Progress (%prog↑), and Number Actions (#act↓). %suc↑ is either 0 or 1 based on the task being completed successfully. %prog↑ is between 0 and 1 indicating progress towards completing the task. #act↓ is the number of actions.

### 5.2 RESULTS AND ANALYSIS

**Overall Results.**

- On the MiniWob++ benchmark, `HeaP Few-shot` matches or outperforms priors works with orders of magnitude fewer demonstrations (21 demos for `HeaP` vs 347k demos in (Furuta et al., 2023) or 2.4M demos in (Humphreys et al., 2022)). See Table 1.
- On the WebArena benchmark (Gitlab, Maps), `HeaP Few-shot` achieves much better success rates than prior works (Zhou et al., 2023; Yao et al., 2022b) that use `Flat` chain-of-thought prompting. See Fig. 4.
- On airline CRM and live websites, we see a similar trend where `HeaP Few-shot` achieves better success rates and task progress with lower number of actions. See Fig. 5, Fig. 7.
- `HeaP Few-shot` achieves higher success rates by breaking down complex tasks into reusable low-level policy calls each of which can be covered with their own in-context examples. See Fig. 2 for an ablation and Figs. 8,9 for qualitative examples.
- Finally, we provide ablations on different model scales and CoT reasoning in Appendix B.

**Comparison to prior works.** In Table 1, `HeaP Few-shot` outperforms or matches priors works with orders of magnitude lower demonstrations on the MiniWob++ benchmark. `HeaP` has an average success rate of 0.96 using only 21 in-context examples.

---

[2]https://platform.openai.com/docs/models

HeaP outperforms all the supervised learning baselines and matches the most perfomant baseline CC-Net (Humphreys et al., 2022) that trains an RL agent using 2.4 million demonstrations. HeaP outperforms the most recent baseline, WebGUM (Furuta et al., 2023) which fine tunes a pre-trained instruction model on 347K demonstrations.

| Method | Models | Training Size | Success Rate |
|---|---|---|---|
| WGE (Liu et al., 2018) | - | 12K+ | 0.77 |
| CC-Net (SL) (Humphreys et al., 2022) | ResNet | 2.4M | 0.33 |
| CC-Net (SL+RL) (Humphreys et al., 2022) | ResNet | 2.4M | 0.96 |
| WebN-T5 (Gur et al., 2022) | T5-XL | 12K | 0.56 |
| WebGUM (HTML) (Furuta et al., 2023) | Flan-T5-XL | 347K | 0.90 |
| Flat / ReAct (Yao et al., 2022b) | GPT-text-davinci-003 | 7 | 0.72 |
| HeaP (Ours) | GPT-text-davinci-003 | 21 | 0.96 |

Table 1: **Comparison to prior works** with success rates averaged across 45 MiniWoB++ tasks. HeaP achieves a higher success rate with orders of magnitude lower samples. See Appendix B.3 for breakup over individual tasks.

Part of the performance gain comes from in-context learning and CoT reasoning with large-scale models similar to ReAct (Yao et al., 2022b). However, HeaP with its hierarchical prompting improves success rates significantly over ReAct (aka Flat), by breaking down complex tasks and covering them efficiently with more in-context examples.

**Why does hierarchical prompting help?**

The key benefit of hierarchical prompting is to break down complex tasks into a set of smaller policies, each of which can be covered by a handful of demonstrations. In contrast, covering the entire task would require combinatorially many more demonstrations. Fig. 2 shows an ablation of HeaP vs Flat with varying number of in-context examples. Hierarchy helps at two levels: (1) For the same number of examples ($\leq 7$), improvements come from decomposing task instructions into granular policy instructions (2) Hierarchical decomposition results in smaller individual policies. This allows us to add more in-context examples ($> 7$) in each policy prompt compared to what is possible with a single flat prompt (see Sec 3) resulting in higher success rates.

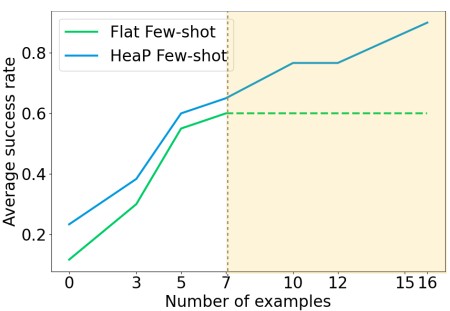

Figure 2: HeaP vs Flat with varying in-context examples on subset of MiniWob++.

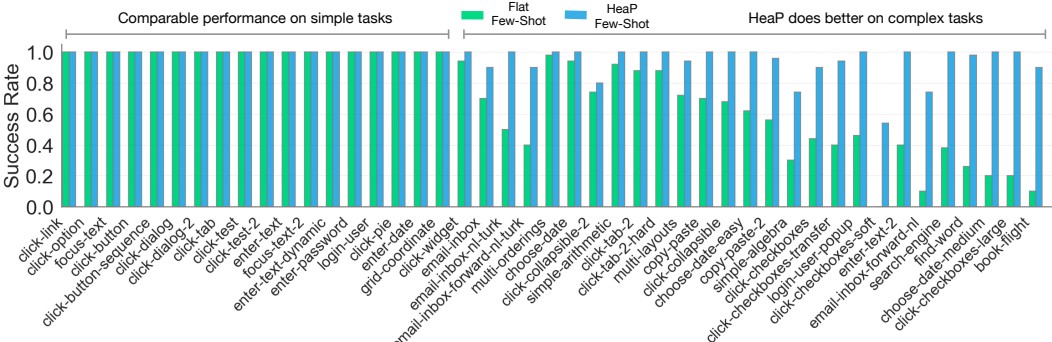

Figure 3: Task-wise success rates breakdown on MiniWob++ (averaged over 50 seeds per task)

We see quantitative evidence for this in Fig. 3 which shows a task-wise success rates breakdown on MiniWob++. The gap between HeaP Few-Shot and Flat Few-Shot is much higher on complex tasks. We characterize complex tasks as those that either require heterogeneous sets of actions or multiple steps with changing webpages. We dive deeper into the book-flight task in Fig. 8 where HeaP Few-shot significantly outperforms baselines. HeaP task planner breaks down the task into a set of policy calls like FILL_TEXT, CHOOSE_DATE. The policies, e.g. CHOOSE_DATE issues a set of low-level actions like CLICK to solve sub-tasks like navigating and picking the right date from a datepicker. This step is particularly challenging for baselines due to the variations in navigating the datepicker. However, the CHOOSE_DATE policy in HeaP Few-shot has the ability to cover these variations with more in-context examples, making it more robust.

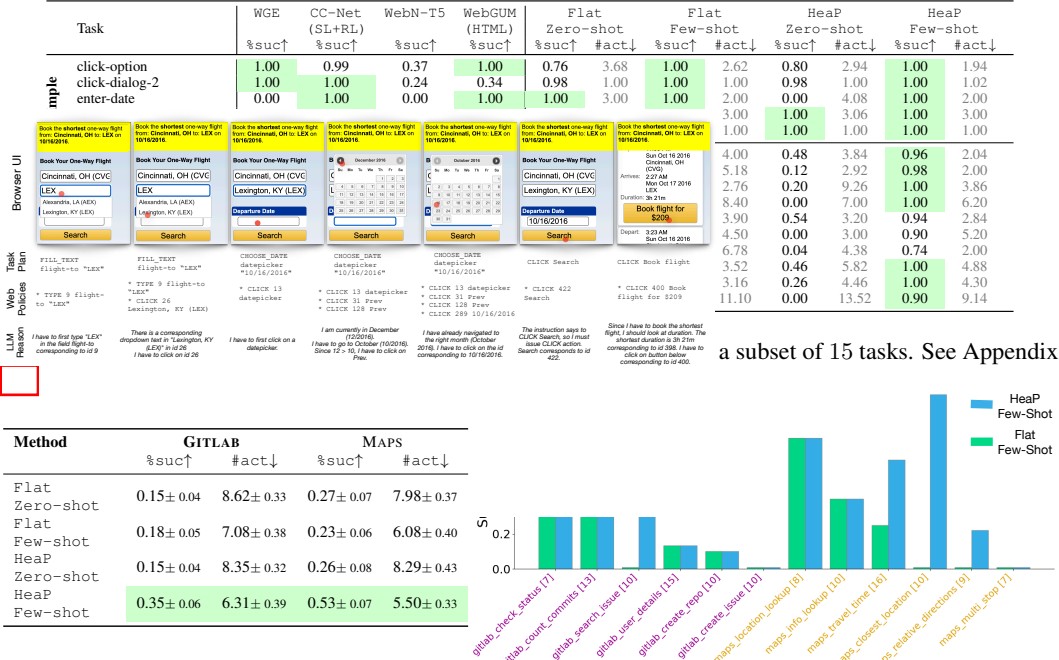

a subset of 15 tasks. See Appendix

| Method | GITLAB %suc↑ | #act↓ | MAPS %suc↑ | #act↓ |
|---|---|---|---|---|
| Flat Zero-shot | 0.15±0.04 | 8.62±0.33 | 0.27±0.07 | 7.98±0.37 |
| Flat Few-shot | 0.18±0.05 | 7.08±0.38 | 0.23±0.06 | 6.08±0.40 |
| HeaP Zero-shot | 0.15±0.04 | 8.35±0.32 | 0.26±0.08 | 8.29±0.43 |
| HeaP Few-shot | 0.35±0.06 | 6.31±0.39 | 0.53±0.07 | 5.50±0.33 |

Figure 4: Evaluation on WebArena Benchmark (Gitlab, Maps). **(Left)** Aggregate metrics **(Right)** Success rate breakdown across 12 intent buckets. Flat Few-Shot is the baseline reasoning agent from WebArena (Zhou et al., 2023) that follows ReAct (Yao et al., 2022b) style of CoT prompting.

| Task | Metric | Flat Zero-shot | Flat Few-shot | HeaP Zero-shot | HeaP Few-shot |
|---|---|---|---|---|---|
| CANCEL FLIGHT | %suc↑ | 0.10 ± 0.10 | 1.00 ± 0.00 | 0.20 ± 0.10 | 1.00 ± 0.00 |
| | %prog↑ | 0.15 ± 0.10 | 1.00 ± 0.00 | 0.80 ± 0.10 | 1.00 ± 0.00 |
| | #act↓ | 11.20 ± 0.20 | 6.00 ± 0.00 | 11.33 ± 0.85 | 6.00 ± 0.00 |
| FIND BOOKING | %suc↑ | 0.00 ± 0.00 | 0.90 ± 0.05 | 1.00 ± 0.00 | 1.00 ± 0.00 |
| | %prog↑ | 0.00 ± 0.00 | 0.90 ± 0.10 | 1.00 ± 0.00 | 1.00 ± 0.00 |
| | #act↓ | 11.00 ± 0.00 | 4.10 ± 0.75 | 3.00 ± 0.00 | 3.00 ± 0.00 |
| SEARCH FLIGHT | %suc↑ | 0.00 ± 0.00 | 0.00 ± 0.00 | 0.00 ± 0.00 | 1.00 ± 0.00 |
| | %prog↑ | 0.50 ± 0.04 | 0.60 ± 0.00 | 0.60 ± 0.05 | 1.00 ± 0.00 |
| | #act↓ | 11.00 ± 0.00 | 11.00 ± 0.00 | 11.00 ± 0.00 | 5.00 ± 0.00 |
| UPDATE PASSENGER DETAILS | %suc↑ | 0.00 ± 0.00 | 0.50 ± 0.15 | 0.30 ± 0.10 | 0.65 ± 0.15 |
| | %prog↑ | 0.00 ± 0.00 | 0.90 ± 0.00 | 0.60 ± 0.10 | 0.90 ± 0.05 |
| | #act↓ | 16.00 ± 0.00 | 11.10 ± 1.25 | 14.30 ± 0.70 | 11.85 ± 0.75 |
| BOOK FLIGHT | %suc↑ | 0.00 ± 0.00 | 0.00 ± 0.00 | 0.00 ± 0.00 | 0.65 ± 0.15 |
| | %prog↑ | 0.55 ± 0.05 | 0.40 ± 0.05 | 0.40 ± 0.05 | 0.82 ± 0.05 |
| | #act↓ | 26.00 ± 0.00 | 26.00 ± 0.00 | 25.75 ± 0.25 | 22.25 ± 0.90 |

Figure 5: **(Left)** Evaluation on 5 airline CRM tasks averaged over 20 randomized scenarios per task. **(Right)** Simulator visualization of a book-flight task consisting of >20 steps.

On the WebArena benchmark, we observe a similar trend in Fig. 4 showing a breakdown of success rates across 12 different intents on 2 domains. Compared to MiniWob++, this is a significantly more challenging environment where prior work with Flat CoT prompting (Zhou et al., 2023; Yao et al., 2022b) has very limited success rates ($\sim 18\%$). This limitation arises from the challenge of understanding how to interact appropriately with web pages. HeaP provides a mechanism for defining dedicated policies that can be taught with targeted in-context examples. For instance, a task like searching a Gitlab issue involves filtering and sorting by specific criteria. A dedicated policy like SEARCH_ISSUE() can handle this by filtering by keywords, sorting, and determining issue status.

**How well does HeaP generalize across tasks?** Table 2 along with Appendix B.3 shows metrics across 45 tasks from MiniWoB++ (Liu et al., 2018; Shi et al., 2017) averaged over 50 seeds per task. HeaP Few-shot obtains higher success rates with lower number of actions compared to baselines, with the performance gap higher for complex tasks, with complex being tasks that either require a heterogeneous set of actions or multiple steps with changing webpages. HeaP Few-shot achieves this with only 21 examples from 6 tasks and is able to solve the remaining 39 tasks without ever having seen them. Table 3 shows the breakup of in-context examples across different environments.

Similarly, Fig. 5 shows metrics on 5 longer horizon CRM tasks (each averaged over 20 scenarios) corresponding to typical airline workflows like find & cancel bookings, update passenger details, find & book flights. `HeaP Few-shot` obtains higher success and task progress with lower number of actions compared to baselines. It achieves this with 10 in-context examples from 2 tasks (Table 3)

**How well does `HeaP` generalize across complex webpages?** Fig. 7 shows evaluation of `HeaP Few-shot` and `Flat Few-shot` across 10 runs each on 3 different live websites with task specification coming from short simulated conversations. What makes this task challenging is that the browser content from these websites have a lot of extraneous information that make it challenging to parse the correct fields. Fig. 6 shows the extent of compression we perform to fit the browser content into the LLM's context space (see Appendix F for details). For WebArena, we use the accessibility tree browser content representation from the environment. For each run, we evaluate by comparing model performance against a reference human demonstration. In Fig. 7, `HeaP Few-shot` is able to generalize to multiple websites even though it has demonstration from only one (i.e. jetblue.com). In contrast, `Flat Few-shot` fails to generalize from it's demonstration. Again `HeaP Few-shot`, by hierarchically decomposing the problem, is able to use demonstrations more efficiently.

**Ablations on reasoning, models, and few-shot examples.** Appendix B shows ablations on CoT reasoning and model scales. Overall, we find CoT to boost performance across tasks, especially multi-step tasks. For models, `gpt-4` improves performance across methods, but having both hierarchical prompting and few-shot ex-

| Environment | Method | Examples | Tasks covered by examples |
|---|---|---|---|
| MiniWob++ | Flat | 7 | choose-date, book-flight |
| | HeaP | 21 | |
| | ⊢ TASK_PLANNER | 3 | |
| | ⊢ FILL_TEXT | 5 | choose-date, book-flight |
| | ⊢ CHOOSE_DATE | 4 | search-engine, click-tab-2 |
| | ⊢ SEARCH_LINK | 3 | click-checkbox, email-inbox |
| | ⊢ SEARCH_TAB | 1 | |
| | ⊢ CLICK_CHECKBOX | 2 | |
| | ⊢ PROCESS_EMAIL | 3 | |
| WebArena | Flat | 3 | count_commits, closest_location, |
| | HeaP | 15 | |
| | ⊢ TASK_PLANNER | 3 | count_commits, |
| | ⊢ FIND_COMMIT | 2 | search_issue, travel_time, |
| | ⊢ SEARCH_ISSUE | 3 | closest_location |
| | ⊢ FIND_DIRECTIONS | 4 | |
| | ⊢ SEARCH_NEAREST_PLACE | 4 | |
| Airline CRM | Flat | 5 | cancel flight |
| | HeaP | 10 | |
| | ⊢ TASK_PLANNER | 3 | |
| | ⊢ FILL_TEXT | 2 | cancel flight, book flight |
| | ⊢ CHOOSE_DATE | 2 | |
| | ⊢ SELECT_FLIGHT | 3 | |
| LiveWeb | Flat | 3 | jetblue.com |
| | HeaP | 5 | |
| | ⊢ TASK_PLANNER | 1 | jetblue.com |
| | ⊢ FILL_TEXT | 2 | |
| | ⊢ CHOOSE_DATE | 2 | |

Table 3: In-context examples for `HeaP` and `Flat`. Each example is a state-action pair at particular timestep.

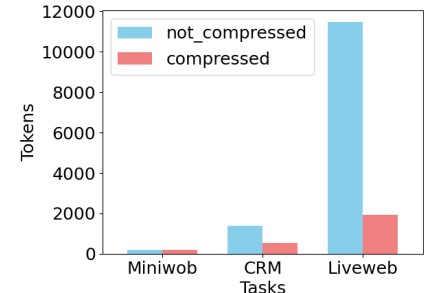

Figure 6: Token counts for browser content before and after compression on different environments.

amples continue to help. `gpt-3.5-turbo` does better in zero-shot setting but under-performs `text-davinci-003` when given few-shot examples. Fig. 9 shows the effect of few-shot examples qualitatively on a search-engine task. Few-shot examples help ground the task in concrete low-level actions on the web UI, resulting in `HeaP Few-shot` navigating to the desired link correctly.

**Error Analysis.** We cluster common failure modes of `HeaP`: (1) *Content parsing errors:* Browser content may be parsed with incorrect associations. Specifically, since we flatten the DOM structure and add to the LLM context, this can cause incorrect text associations. (2) *Error recovery:* LLM may not know how to recover from incorrect actions. For instance, `HeaP` clicks on a wrong link,

| | Metric | Flat Few-shot | HeaP Few-shot |
|---|---|---|---|
| jetblue.com | %suc↑ | 0.60± 0.05 | 1.00± 0.00 |
| | %prog↑ | 0.75± 0.02 | 1.00± 0.00 |
| | #act↓ | 8.00± 0.00 | 7.00± 0.00 |
| united.com | %suc↑ | 0.20± 0.04 | 0.50± 0.17 |
| | %prog↑ | 0.47± 0.03 | 0.68± 0.11 |
| | #act↓ | 8.00± 0.00 | 6.40± 0.27 |
| aa.com | %suc↑ | 0.00± 0.00 | 0.20± 0.13 |
| | %prog↑ | 0.40± 0.04 | 0.60± 0.08 |
| | #act↓ | 6.00± 0.00 | 6.00± 0.00 |

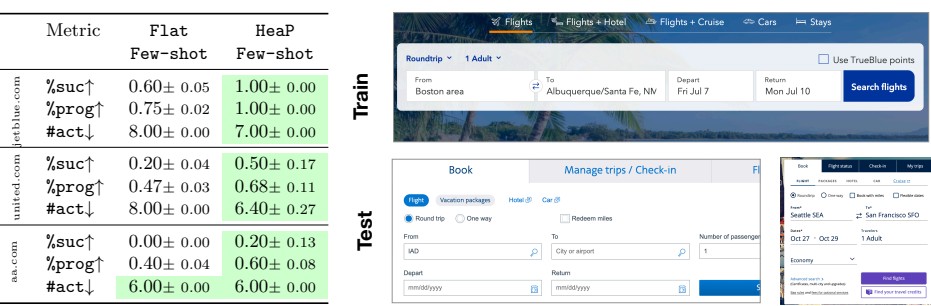

Figure 7: **(Left)** Evaluation on 3 live airline websites averaged over 10 runs per website. **(Right)** Difference in train (jetblue) v/s test (united, aa) website UIs.

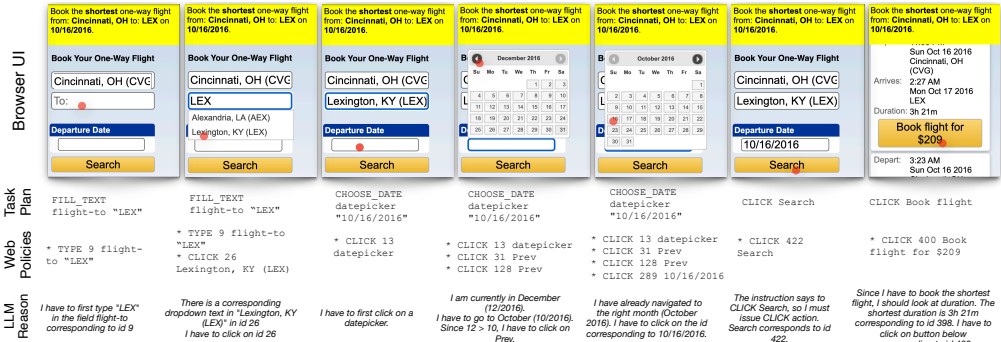

Figure 8: Outputs from `HeaP Few-shot` on book-flight task showing hierarchical task planner actions, low-level web policy actions, and LLM reasoning.

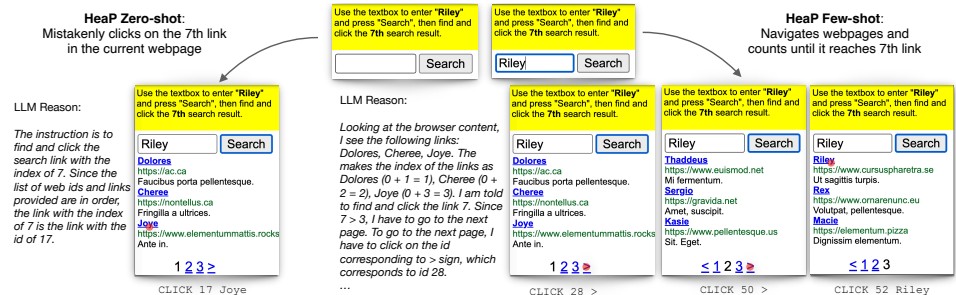

Figure 9: `HeaP Few-shot` vs `HeaP Zero-shot` on a search-engine task. The instruction asks to find the 7th link, however, it is ambiguous what 7 refers to. `HeaP Few-shot` with a single in-context demo is able to ground the task in the UI and reason that the 7th link lies in the 2nd webpage and navigates to the link correctly.

sending it to a new webpage not seen in the demonstrations. (3) *Visual information gaps:* Visual elements, such as specific dropdown menus in maps environment, do not appear in the DOM. Such tasks require multi-modal models that reason about browser images.

## 6 DISCUSSION AND LIMITATIONS

In this paper, we present a hierarchical framework `HeaP` for solving web tasks from few-shot demonstrations. We evaluate against a range of baselines on a suite of web tasks and characterize performance gains from both hierarchical prompting and demonstrations. Our key takeaways are:

**(1) Hierarchy breaks down complex tasks** Our results indicate that hierarchical prompting achieves higher success rates by breaking down complex tasks into reusable low-level policy calls (see Fig. 9). This is evident in the performance difference between `HeaP Few-shot` and `Flat Few-shot` (see Figs. 3,4,5,7), with Fig. 2 showing the role of hierarchy in both better task decomposition and ability to pack in more examples. **(2) Sample efficient generalization** `HeaP` matches or outperforms priors works with multiple orders of magnitude less data (see Table 1). It is able to adapt to unseen tasks with only a handful of task demonstrations seen in-context (see Table 3). **(3) Effects of few-shot prompting and reasoning** Few-shot examples in the prompt are effective at grounding high-level task instructions as actions on the web UI environment (see Fig. 9). CoT reasoning significantly boosts performances across all methods, particularly on multi-step tasks (see Appendix B).

While `HeaP` shows promise, there are still limitations and open challenges: **(1) Complex Webpages.** `HeaP` is currently unable to handle pages with visual only components since those observations don't get parsed from the HTML DOM. Leveraging pretrained multi-modal models offer a promising avenue (Lee et al., 2023; Furuta et al., 2023). Moreover, parsing pages containing long tables, databases needs advanced compression techniques such as learning dedicated saliency models (Wang et al., 2022; Sridhar et al., 2023) to determine relevant web elements. **(2) Verification and Error Recovery.** `HeaP` may click on a wrong link sending it to a new webpage and must learn to recover from such errors. Learning from incorrect actions either via human feedback or self-verification are interesting directions of future work. Action LLMs also carry potential for misuse given their execution on open-domain environments, requiring careful verification and security solutions.

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
