# OpenReview forum: "HeaP: Hierarchical Policies for Web Actions using LLMs"
_ICLR.cc/2024/Conference — Submitted to ICLR 2024_

### Official Review · Reviewer_NxCH · 2023-10-19

**Soundness:** 3 good
**Presentation:** 4 excellent
**Contribution:** 2 fair
**Rating:** 3
**Confidence:** 4

**Summary:**

The work presents their design of a web action agent with LLM that can follow the natural language instruction and perform actions on the website. The agent consists of a high-level planner and a low-level actor, both of which invoke LLM for concrete output given different input context and prompt. The core idea is to decompose the task to achieve higher performance and generalization capacity.

**Strengths:**

The design of this method is sound and reasonable.

Exhaustive details of the prompt and results analysis are presented.

**Weaknesses:**

The evaluation is based on too few samples: 45 tasks on MIniWob++, 125 examples of two domains on WebArena, 5 distinct tasks with 20 scenarios on Ariline CRM, and 3 website with 10 searches per site on Live Websites.

Given the fact that human demonstration is collected to form the prompts to the LLM in HEAP, it should be actually evaluated on more diverse websites instead of fewer websites.

**Questions:**

1. Could the author elaborate on the demonstration collection process described in 4.2?

2. What's the purpose of D_{label} on the end of page 4 (Sec 4.2)? How does this dataset utilized in the HEAP?

3. What is the trainable component in the HEAP? e.g. which function / parameters described in Algorithm 1 is learnable?

4. Why the models are not evaluated on the entire benchmarks but only a set of them.

5. What does training size 21 in Table 1 last row mean? The HEAP was shown 21 samples, in which format, to train which part?

6. It really depends on the demonstration collected and the diversity of evaluation cases that whether the benefit claim of "sample efficient" and "generalization" are sound.

---

> ### Author Response · Authors · 2023-11-15
>
> We thank the reviewer for their feedback! Please find below our response:
>
> **The evaluation is based on too few samples: 45 tasks on MIniWob++, 125 examples of two domains on WebArena, 5 distinct tasks with 20 scenarios on Ariline CRM, and 3 website with 10 searches per site on Live Websites.**
>
> * We provide the most comprehensive evaluation on web tasks, i.e. 4 distinct environments including live websites and totaling ~2500 evaluations. Prior works in this domain [1-5] (in Table 1) evaluate on 1-3 datasets, and none evaluate on live websites.
> * On MiniWoB++, we evaluate all tasks that do not require visual inputs (see Table 4 in Appendix B for detailed breakdown). Each task was evaluated on 50 seeds totalling 2250 (=45*50) episodes or trajectories lasting 2-10 timesteps.
> * We evaluate two distinct domains on WebArena, a SOTA challenging dataset that came out only a couple months ago. We outperform the best performing LLM agent in their paper by a factor of 2x by leveraging hierarchy (see Fig. 4). None of the prior works in Table 1 evaluate on WebArena. In fact, since prior works use large amounts of demonstrations (e.g. 347k, 12k, 2.2M), it’s not possible to use those on WebArena tasks since they don’t have large-scale human demonstrations yet.
> * We can certainly evaluate up to 100 scenarios on Airline CRM, but the confidence intervals are fairly tight.
>
>
> **Given the fact that human demonstration is collected to form the prompts to the LLM in HEAP, it should be actually evaluated on more diverse websites instead of fewer websites.**
> * We reiterate that to the best of our knowledge is this is the most diverse evaluation on web automation tasks compared to prior works [1-5] in this domain.
> * Live websites are challenging since web pages are complex containing ~12k tokens, that we compress down to 2k tokens in text formats for an LLM without loss of performance (see Fig. 6). The purpose of showing live website evaluations is to show the real-world applicability of our method.
> * It is not feasible to do extensive evaluation on the live websites (none of the prior works [1-5] in this domain do this either):
>     * Bot detectors
>     * Safety constraints
>     * Evaluating success
>     * Require visual reasoning that needs multi-modal models
>
> **Could the author elaborate on the demonstration collection process described in 4.2?**
> * Following is the excerpt from Appendix D.1 where we detail this
>     * We collect demonstrations from a human user who is given a conversation context \phi and performs a sequence of actions in the browser to complete the task.
>     * We design a browser plugin to record webpage DOM d and events such as clicks and types.
>     * Each demonstration is parsed as text by converting the DOM tree into a list of salient web elements like links, buttons, inputs. Each element is represented as text, e.g. <button id=18 title="Travelers">1 Adult />. Action events are parsed as a CLICK <id> or TYPE <id> <val>.
>
> **What's the purpose of D_{label} on the end of page 4 (Sec 4.2)? How does this dataset utilized in the HEAP?**
> * D_label is used to generate examples for both high-level and low-level prompts.
> * Following is the excerpt from Appendix D.2 where we detail this
>     * Once we have a dataset of labeled demonstrations D_{label} we can use that to generate base prompts for both the high-level task planner and low-level policies.
>     * For the task planner, we concatenate the following input–output pairs as in-context examples in the base prompt: {input: [context \phi, initial state s0]} and {output: sequence of web policy calls (\pi1, \psi1), (\pi2, \psi2) . . . , (\piT , \psiT )}. For each web policy, we search the dataset D_label for all instances of the policy and create a prompt by concatenating examples of {input: [instruction \psi_t, current state s_t, previous actions a_{t:t−k} ]} and {output: next action a_t}.
>
> **What is the trainable component in the HEAP? e.g. which function / parameters described in Algorithm 1 is learnable?**
> * Since HEAP uses in-context learning, the parameter that can be varied is the input prompt (P_task in Line 7, P_policy in Line 12), specifically the few-shot examples in the prompt.
>
> **Why the models are not evaluated on the entire benchmarks but only a set of them.**
> * On MiniWoB++, we evaluate all tasks that do not require visual inputs.
> * WebArena is a recent and challenging benchmark, and we picked two domains that were easy to collect demonstrations and did not require visual inputs.
>
> **What does training size 21 in Table 1 last row mean? The HEAP was shown 21 samples, in which format, to train which part?**
> * Please see Table 3 which shows the breakup of the 21 examples in HeaP by examples per policy (see row MiniWob++).
> * Each example is a tuple of (context, browser content, previous action, action). The examples are used as part of the prompt that is sent to an LLM to predict the action. See Appendix G.1, Listing 2 for the final prompt with examples.

---

> > ### Author Response · Authors · 2023-11-15
> >
> > (cont'd)
> >
> > **It really depends on the demonstration collected and the diversity of evaluation cases that whether the benefit claim of "sample efficient" and "generalization" are sound.**
> > * We consider all tasks from MiniWob++ that don’t require visual reasoning and compare results against prior works [1-5]
> > * Our demonstrations are collected in the same style as prior works, i.e. a human doing a task on the web.
> > * Prior works required 12k, 347k, 2.4M demonstrations. We match or outperform with only 21 demonstrations (see Table 1).
> >
> >
> > [1] Liu et al.. Reinforcement learning on web interfaces using workflow-guided exploration, ICLR 2018. https://arxiv.org/abs/1802.08802.
> >
> > [2] Humphreys et al., A data-driven approach for learning to control computers, ICML 2022. https://arxiv.org/abs/2202.08137
> >
> > [3] Gur et al., Understanding HTML with large language models, arXiv 2022 https://arxiv.org/abs/2210.03945
> >
> > [4] Furuta et al., Multimodal web navigation with instruction fine tuned foundation models. arXiv, 2023. https://arxiv.org/abs/2305.11854
> >
> > [5] Yao et al.. React: Synergizing reasoning and acting in language models. ICLR 2023. https://arxiv.org/abs/2210.03629

---

### Official Review · Reviewer_4BRb · 2023-10-29

**Soundness:** 3 good
**Presentation:** 2 fair
**Contribution:** 3 good
**Rating:** 6
**Confidence:** 2

**Summary:**

This paper introduces a framework that learns hierarchical prompts from demonstrations for planning high-level tasks and executing them via a sequence of low-level policies. The approach decomposes complex web tasks into a sequence of high-level subtasks, each of which can be then solved by a sequence of low-level policies. This method learns hierarchical LLM prompts for both levels of tasks and policies. The approach was evaluated on a range of increasingly complex benchmarks and the results show that the proposed approach achieves excellent performance compared with existing approaches.

**Strengths:**

The approach introduces a novel hierarchical approach to prompt LLMs to perform web tasks. Experimental results on various complex web benchmarking datasets show the superiority of the proposed approach.

**Weaknesses:**

I recommend moving some implementation details like prompts into the main body to help the reader better understand the work.

**Questions:**

Do you label the high-level task plans?

---

> ### Author Response · Authors · 2023-11-15
>
> We thank the reviewer for their feedback! Please find below our response:
>
> **I recommend moving some implementation details like prompts into the main body to help the reader better understand the work.**
> * We are happy to do so
>
> **Do you label the high-level task plans?**
> * We do not. The demonstrations collected from the human demonstrations are states and low-level actions. We then use “autolabeling” at each timestep to label the low-level policy that was being executed. Our scripts convert the labeled datasets to prompts for both high-level planners and low-level policies. Finally, we augment prompts with chain-of-thought reasoning.

---

### Official Review · Reviewer_oBNG · 2023-10-31

**Soundness:** 2 fair
**Presentation:** 3 good
**Contribution:** 2 fair
**Rating:** 3
**Confidence:** 4

**Summary:**

This paper presents a method HeaP for LLM to perform tasks on the web. It first asks LLM to decompose the task into several steps as a planner, where each step is a call to low-level policies. Then within each low-level policy, LLM is called to predict the next action sequentially. Both the planner and low-level policy execution have prompts constructed automatically by collecting few shot examples from autolabeled human demonstrations. Extensive experiments over several datasets demonstrate the gain over ReAct which does not use hierarchical planning.

**Strengths:**

- The paper is overall easy to read, although some important methodological details like autolabeling and prompt construction are in appendix which makes it hard to read.
- The experiments are extensive over 4 datasets with many tasks. The gain demonstrated is substantial.

**Weaknesses:**

- The idea of hierarchical planning with a high-level planner and low-level policies using LLM has been explored by many previous robotics works e.g. LLM-planner (https://arxiv.org/pdf/2212.04088.pdf and the line of works they cited). Additionally, PaP (https://aclanthology.org/2022.suki-1.8.pdf) and Parsel (https://arxiv.org/pdf/2212.10561.pdf) have also explored similar ideas of prompting LLM to generate a hierarchical plan but implementing low-level planners with programs. Implementing both high-level and low-level planners with LLM prompting has been explored in Decomposed Prompting (https://openreview.net/pdf?id=_nGgzQjzaRy). Considering these previous works, the novelty of this paper is limited to applying existing ideas to web datasets and potentially the technical details of autolabeling from human demonstrations.
- The low-level policies are manually defined during autolabeling, making the framework limited in flexibility comparing to previous works that allow LLM to generate decompositions freely.
- The only LLM prompting baseline compared against is ReAct, which demonstrates the benefits of hierarchical planning. However, such benefits have been demonstrated with the prior works mentioned above.

**Questions:**

n/a

---

> ### Author Response · Authors · 2023-11-15
>
> We thank the reviewer for the feedback! Please find below our responses:
>
> **The idea of hierarchical planning with a high-level planner and low-level policies using LLM has been explored by many previous robotics works**
>
> There seems to be misunderstanding on the claimed contributions of this paper.
> * We agree that hierarchical planning is a fundamental concept in AI and has been the bedrock of robotics and decision making at large.
> * We don’t make any universal claims on hierarchical planning with LLMs
>     * Although we note that at the time of writing this paper there is no work (to the best of our knowledge) that uses LLMs for both high and low-level  planning
> * Instead our focus is on enabling LLMs to solve tasks on the web
>    * This is an increasingly important domain, where we present the first results from an LLM based approach that achieves SOTA results with orders of magnitude lower demonstrations (see Table 1). We do so by showing that hierarchical policies are essential for solving web tasks.
> We provide the most comprehensive evaluation on web tasks, i.e. 4 distinct environments including live websites and totaling ~2500 evaluations. Prior works in this domain [1-5] (in Table 1) evaluate on 1-3 datasets and none evaluate on live websites.
>
> We are happy to update the writeup to clear any misunderstanding.
>
> **Considering these previous works, the novelty of this paper is limited to applying existing ideas to web datasets and potentially the technical details of autolabeling from human demonstrations.**
>
> Thank you for the related works, we will include the relevant ones in the paper. However, we note that:
> * None of them propose hierarchical LLM policies (LLM is only used as the high-level planner)
> * None of the them look at the web domain
> * Some of them don’t even look at planning tasks (e.g. Q&A)
> * None of them propose a method to auto-generate prompts by autolabeling demonstrations
>
> Going into further details
> * LLM-planner https://arxiv.org/pdf/2212.04088.pdf
>     * Focuses exclusively on vision-language navigation, a very different domain
>     * Moreover, only the high-level planner is LLM, low-level planner is a combination of engineered/learned components
>     * The authors themselves mention that none of the cited works use LLMs for hierarchical planning in the VLN domain.
> * Paap https://aclanthology.org/2022.suki-1.8.pdf
>     * Focuses on defining a hierarchical grammar. Doesn’t use LLMs or train a model.
> * Parsel https://arxiv.org/pdf/2212.10561.pdf
>     * Uses a LLM to decompose a natural task into a bunch of subtasks that is then fed to a codeLM to generate a code
>     * There is no policy / feedback from the real world
> * Decomposed prompting https://openreview.net/pdf?id=_nGgzQjzaRy
>     * Does not look at planning tasks but rather question answering
>     * Requires users to manually annotate low-level policies while we autolabel
>
> **Why is the web simply not another dataset?**
>
> * Web is an increasingly important and challenging domain for decision making
> * Many fundamentals challenges to creating web agents:
>     * Countless ways to interact with the web, leading to a combinatorially large number of tasks.
>     * Web interfaces differ from one website to another. It is intractable to cover all such variations in the prompt.
> * We achieve SOTA results compared to prior web automation works and do this with orders of magnitude less data (see Table 1). To achieve this, we had to solve a number of problems:
>     * Intractable to cover all tasks and web interface combinations in a single prompt, that we tackle with our hierarchical LLM approach, HeaP
>     * Tedious for users to segment demonstrations into low-level policies, that we automate via autolabellng (see Appx D)
>     * Complex web pages that can have up to 12k tokens, that we compress to 2k without loss of performance (see Fig. 6)
>
> [1] Liu et al.. Reinforcement learning on web interfaces using workflow-guided exploration, ICLR 2018. https://arxiv.org/abs/1802.08802
>
> [2] Humphreys et al., A data-driven approach for learning to control computers, ICML 2022. https://arxiv.org/abs/2202.08137
>
> [3] Gur et al., Understanding HTML with large language models, arXiv 2023 https://arxiv.org/abs/2210.03945
>
> [4] Furuta et al., Multimodal web navigation with instruction fine tuned foundation models. arXiv, 2023. https://arxiv.org/abs/2305.11854
>
> [5] Yao et al.. React: Synergizing reasoning and acting in language models. ICLR 2023. https://arxiv.org/abs/2210.03629

---

> > ### Author Response · Authors · 2023-11-15
> >
> > (cont'd)
> >
> > **The low-level policies are manually defined during autolabeling, making the framework limited in flexibility compared to previous works that allow LLM to generate decompositions freely.**
> >
> > * We don’t find this to be a problem since most tasks can be decomposed into smaller and relatively easy-to-identify subtasks.
> > * The only manual effort is providing one demonstration per subtask (low-level policy), and the autolabelling can use that to label un-annotated demonstrations
> > * It might be true that an LLM can be used to automatically decide the granularity, we are to happy to include references especially on web tasks that the reviewer might have in mind
> >
> >
> > **The only LLM prompting baseline compared against is ReAct, which demonstrates the benefits of hierarchical planning. However, such benefits have been demonstrated with the prior works mentioned above.**
> >
> > * We reiterate that our focus is on enabling LLMs to solve tasks on the web
> > *  No prior work has shown in-context learning results on web benchmarks like MiniWob++ or demonstrated hierarchical LLM policies vs ReAct
> > * It would be interesting to compare HeaP vs ReAcT on robotic tasks, but this is out of scope for this paper that focuses on web tasks

---

> > > ### Comment · Reviewer_oBNG · 2023-11-22
> > >
> > > Thank you for the detailed response. I will maintain the current score.

---

### Official Review · Reviewer_aRc1 · 2023-11-06

**Soundness:** 3 good
**Presentation:** 3 good
**Contribution:** 3 good
**Rating:** 8
**Confidence:** 3

**Summary:**

The paper introduces HeaP, a framework that leverages Large Language Models (LLMs) to decompose complex web tasks into modular sub-tasks. It uses a hierarchical approach, learning high-level task plans and low-level policies from human demonstrations, enabling LLMs to perform web actions effectively. It addresses challenges related to the combinatorially large space of web tasks and variations in web interfaces. Experimental results demonstrate that HeaP outperforms previous methods with significantly fewer training examples on various web tasks and interfaces such as MiniWoB++, WebArena, and a mock airline CRM

**Strengths:**

-Originality: The idea is interesting in the way HeaP leverages hierarchical policies to decompose complex web tasks using a high-level task planner
 into modular  low-level web policies.

-Quality: The paper is quite thorough in its experimental setup as it tests on 4 interesting datasets, including simulated and live websites, to assess the performance of the proposed approach.

-Clarity: The paper is well-written and structured, making it easy for readers to follow and understand the proposed approach

-Significance: The paper addresses a significant challenge in the field of natural language processing and machine learning, which is teaching LLMs to perform web-based tasks which can lead to a huge set of applications

**Weaknesses:**

- The tasks are not that challenging and the results are very weak relative to how powerful the LLM model used here which is GPT-3.5. For example, it seems that the proposed method struggles with book-flight which is a basic constrained task and therefore this method is very far from being deployed in the real world

- Using closed source methods like GPT-3.5 is expensive. I'd be curious to see how this method would perform with open source methods like Llama and Mistral.

- No code was provided to asses and verify the results as well as understand the low level details of how the method is implemented

**Questions:**

Please address the weaknesses above.

---

> ### Author Response · Authors · 2023-11-16
>
> We thank the reviewer for their feedback! Please find below our responses:
>
> **The tasks are not that challenging .. seems that the proposed method struggles with book-flight which is a basic constrained task .. method is very far from being deployed in the real world**
>
> _Contextualizing work against existing baselines and benchmarks_
>
> * We rigorously compare our approach against baselines, including prior works and zero vs few-shot and flat vs hierarchy methods. We evaluate tasks across 4 distinct environments including live websites, totaling ~2500 evaluations.
> * We achieve SOTA results compared to prior web automation works [1-5] and do this with two orders of magnitude less data (see Table 1).
> * Live websites are challenging since web pages are complex containing ~12k tokens, that we compress down to 2k tokens in text formats for an LLM without loss of performance (see Fig. 6, Appx F).
> * The purpose of showing live website evaluations is to show the real-world applicability of our method. However, it is not feasible to do extensive evaluation on the live websites (none of the prior works [1-5] do this either) due to:
>    * Bot detectors
>    * Safety constraints
>    * Evaluating success
>    * Require visual reasoning that needs multi-modal models
>
> _On MiniWob++ book-flight task_
>
> * While prior works scored 0.00, 0.87, 0.00, 0.48, 0.00 on this task, our approach gets a success rate of 0.9 (see task-wise performance breakdown in Table 4, Appx B)
> * This task's complexity lies in navigating UI elements such as date grids and applying filters, e.g., "shortest," "one-way vs. round trip," "cheapest", over multiple steps with changing pages (see Fig. 8 for illustration)
> * HeaP handles these challenges through dedicated low-level policy prompts like "choose-date", "fill-text," that get composed given a high-level task like "book-flight"
>
> _On airlinecrm book-flight task_
>
> * Our motivation for building this simulator that we plan to open-source was to simulate longer-horizon tasks with more complex webpages (see Appx E)
> * The book-flight task is connected to a mock database and involves >20 steps (as opposed to ~10 steps in MiniWoB++).
> * The complexity lies in solving more involved subtasks like adding passenger details, payment information, selecting flight schedule while interacting with diverse UI elements like text boxes, selection links, date boxes.
> * We show benefits of both hierarchy and few-shot examples in solving these tasks
>
> _Limitations and Open Challenges_
>
> We acknowledge there are many limitations and open challenges that need to be solved before our approach can be safely and reliably deployed in the real-world on an open set of tasks. We list these in Limitations (Sec 6) and Broader Impacts (Appx A), in particular,
>
> * **Complex webpages.** Parsing pages containing long tables, databases, etc needs advanced compression techniques such as learning dedicated saliency models.
> * **Multimodal observations.** HeaP is currently unable to handle pages with visual only components since those observations don’t get parsed from the HTML DOM. Leveraging pretrained multi-modal models offer a promising avenue.
>
> * **Verification.** Real-world applications would require a verification module to ensure the successful completion of actions, especially in cases where websites may crash or reset. Using LLMs to verify actions is another promising direction of future work.
>
> * **Error Recovery.**  HeaP may click on a wrong link sending it to a new webpage and must learn to recover from such errors. Learning from incorrect actions either via human feedback or self-verification are interesting directions of future work.
>
> * **Safety.** Finally, action LLMs carry potential for misuse given their execution on open-domain environments, requiring careful security and verification solutions.
>
> We are happy to include or expand out any points here that the reviewer thinks are important!
>
> [1] Liu et al.. Reinforcement learning on web interfaces using workflow-guided exploration, ICLR 2018. https://arxiv.org/abs/1802.08802.
>
> [2] Humphreys et al., A data-driven approach for learning to control computers, ICML 2022. https://arxiv.org/abs/2202.08137
>
> [3] Gur et al., Understanding HTML with large language models, arXiv 2022 https://arxiv.org/abs/2210.03945
>
> [4] Furuta et al., Multimodal web navigation with instruction fine tuned foundation models. arXiv, 2023. https://arxiv.org/abs/2305.11854
>
> [5] Yao et al.. React: Synergizing reasoning and acting in language models. ICLR 2023. https://arxiv.org/abs/2210.03629

---

> > ### Author Response · Authors · 2023-11-16
> >
> > (cont'd)
> >
> > **Using closed source methods like GPT-3.5 is expensive. I'd be curious to see how this method would perform with open source methods like Llama and Mistral.**
> >
> > That is an interesting question. We did try few-shot prompting with llama2-7b but did not find the performance to be as good as gpt-{3,3.5,4}. Common failure modes seem to be:
> > * Not following the desired action formats,
> >    * e.g. saying “Click #justo” instead of “Click 4” for a browser content like “… <div id=2 val=wrap />\n<link id=4 val=justo. />...”
> > * Predicting multiple actions at the same time with spurious text in between
> > * Skipping over intermediate actions sometimes,
> >    * e.g. on a book flight task with browser content like “... <h2 id=5 val=Book Your One-Way Flight />\n<input_text id=7 val=flight-from />…”, the model generates "Click 5" directly before filling in the empty flight-from input box.
> >
> > We are working to improve the prompt for llama2, adding in more explicit instructions using the same few-shot examples. We will re-run quantitative MiniWob++ evaluations with that and hope to share results later in the week.
> >
> > **No code was provided to asses and verify the results as well as understand the low level details of how the method is implemented**
> >
> > We plan to release code and data but have not yet been able to due to internal review processes (especially around the web plugin and tooling). We do mention open-sourcing the code in the paper and commit to releasing it prior to the conference.
> >
> > We include the prompts in Appx G and also illustrate the HeaP planner-policy code architecture in Fig. 12, Appx C. We are happy to address any specific questions on the implementation in the meantime.

---

> ### Author Response · Authors · 2023-11-20
> **Llama2 evaluation**
>
> We ran evaluations with the llama2-13b model and include results along with common failure modes as Appendix G in the paper. Overall, we find the performance to be lower than gpt-{3,3.5,4}. The drop in performance could be due to a number of reasons such as the model size or the training data on which the models are trained. However, we find that HeaP still outperforms Flat for many tasks. We also observe that the few-shot examples do not improve performance. We think that fine-tuning the model on a larger dataset of demonstrations can help address some of these failure modes and leave that as interesting future work.

---

> > ### Author Response · Authors · 2023-11-22
> > **Code Release**
> >
> > We were able to get permissions in time and have uploaded the code in the supplementary materials.
> >
> > The HeaP planner policy implementation can be found in src/policy/heap_policy.py. All web environments are under src/environment that wraps around appropriate webpage parsers, src/parsers. The environments follow a similar API as gym environments. All prompts are under src/prompts.
> >
> > We’ve added details to the Readme on how to run the miniwob evaluations. Evaluation metrics and plots can be found under analysis/notebooks.

---

> ### Comment · Reviewer_aRc1 · 2023-12-05
> **Response to Authors**
>
> Thanks for the detailed response and your efforts for the extra Llama experiments. I have increased the score to 8.

---

### Meta-Review · Area_Chair_Z36W · 2023-12-11

**Metareview:**

This paper proposes a novel framework called HeaP, which leverages Language Models (LLMs) to decompose web tasks into sub-tasks and solve them using low-level policies. By learning hierarchical LLM prompts from demonstrations, HeaP enables planning high-level tasks and executing low-level policies, outperforming prior works with significantly less data across various web task benchmarks. The framework demonstrates its efficacy on tasks such as MiniWoB++, WebArena, a mock airline CRM, and live website interactions. There are some weaknesses of the paper raised from the review comments and discussions, including the incremental technical novelty of the hierarchical policies, framework limitation, the lack of compared baselines. Although the authors provided detailed feedbacks, some of the concerns raised are still unsolved.

**Justification For Why Not Higher Score:**

There are some weaknesses of the paper raised from the review comments and discussions, including the incremental technical novelty of the hierarchical policies, framework limitation, the lack of compared baselines. Although the authors provided detailed feedbacks, some of the concerns raised are still unsolved.

**Justification For Why Not Lower Score:**

N/A

---

### Decision · Program_Chairs · 2024-01-16

Reject